# GEASS: Neural causal feature selection for high-dimensional biological data

**Mingze Dong**
Yale University
`mingze.dong@yale.edu`

**Yuval Kluger**
Yale University
`yuval.kluger@yale.edu`

## Abstract

Identifying nonlinear causal relationships in high-dimensional biological data is an important task. However, current neural network based causality detection approaches for such data suffer from poor interpretability and cannot scale well to the high dimensional regime. Here we present GEASS (Granger fEAture Selection of Spatiotemporal data), which identifies sparse Granger causal interacting features of high dimensional spatiotemporal data by a single neural network. GEASS maximizes sparsity-regularized modified transfer entropy with a theoretical guarantee of recovering features with spatial/temporal Granger causal relationships. The sparsity regularization is achieved by a novel combinatorial stochastic gate layer to select sparse non-overlapping feature subsets. We demonstrate the efficacy of GEASS in several synthetic datasets and real biological data from single-cell RNA sequencing and spatial transcriptomics.

## 1 Introduction

Advances in single-cell omics research enable full characterizations of high-dimensional gene dynamics in biological systems on a either temporal or spatial scale. An example for the temporal case is single-cell RNA sequencing (scRNA-seq) trajectories, where cells are sampled from a dynamical biological process, sequenced, and ordered based on either real sampled time or inferred pseudo-time (Cannoodt et al., 2016; Saelens et al., 2019). Gene dynamics along the specified cell order encodes information of causal regulation for the underlying biological process. An example for the spatial case is single-cell level spatial transcriptomics (e.g. SeqFISH+ (Eng et al., 2019), Merfish (Fang et al., 2022)), in which cells from a tissue slice are sequenced with their spatial coordinates preserved (Moses and Pachter, 2022; Rao et al., 2021; Palla et al., 2022). Spatial profiling allows investigations of the cellular interplay, corresponding to conditional gene expression change caused by neighborhood phenotypic states. However, despite the potential significance, data-driven causal discovery for such data remains largely unexplored, especially for the spatial omics data.

Identifications of causal regulatory patterns in such data can be reformulated into the general task of causal feature selection in observational data with intrinsic structures, e.g. spatial data or temporal data. Identifications of causal interactions in time-series has lead to valuable findings in multiple disciplines, including but not limited to, economy, climate science, and biology (Hoover, 2006; Kamiński et al., 2001; Runge et al., 2019a).

Learning directed causal relationships in temporal/spatial data is feasible as time and space both induce asymmetric dependencies. In the case of time-series data, a feature in the future cannot have effect on past values of other features. For spatial data, a similar definition of causal dependency can be established (Herrera Gómez et al., 2014).

The concept of Granger causality is proposed in order to uncover the assymetric causal dependency (Granger, 1969; Shojaie and Fox, 2022). In time-series data, this would translate to identifying one variable's causal relationship with other variables based on how well the historical observations of other variables can predict the variable's present value. The application of Granger causality in a spatial context corresponds to predicting significant relationships between neighboring observations of other variables and the specified variable (Mielke et al., 2020), which is a key insight used in recent works aimed to discover cellular interaction patterns in spatial omics data (Fischer et al., 2021; Valdés-Sosa et al., 18).

In the nonlinear regime, information-theoretic measures such as directed information, transfer entropy (Schreiber, 2000), and partial transfer entropy (Staniek and Lehnertz, 2008), are used as a counterpart of linear Granger causality. Moreover, some works consider modeling conditional independence (CI) in time-series data to identify the underlying causal graph (Entner and Hoyer, 2010; Malinsky and Spirtes, 2018; Moneta et al., 2011; Runge et al., 2019a; Pfister et al., 2019; Mastakouri et al., 2021). Two examples are VarLINGAM (Hyvärinen et al., 2010) and PCMCI (Runge et al., 2019b), which are generalizations of LINGAM (Shimizu et al., 2006) and PC (Spirtes et al., 2000) respectively. Finally, multiple recent works have proposed to use neural network approaches to model the nonlinear Granger causality, including MLP, LSTM, and neural-ODE based approaches, resulting in improved prediction power for nonlinear time-series dynamics (Li et al., 2017; Tank et al., 2021; Nauta et al., 2019; Yin and Barucca, 2022; Bellot et al., 2021).

Despite the success of these methods in various systems of interest, multiple challenges limit their use in high-dimensional biological datasets.

- Although linear methods (LINGAM, linear Granger causality) have succeeded in various settings and can potentially scale to high feature numbers, these methods may completely fail when the feature dependency in data is highly complex and nonlinear.

- As the number of conditional independencies generally scales exponentially or at least polynomially with the feature size, applying causal discovery methods which are based on CI tests to high-dimensional data is not realistic. Distinctively, Granger-causality based methods are built with a prediction model for each feature in the data. The time complexity of solving the stacked prediction model for all features is of polynomial level with respect to the feature size.

- In previous methods, the number of causal edges between features is assumed to be sparse (edge sparsity) to maximize interpretability of the identified causal graph. However, in biological data, there exists a large proportion of nuisance features. Also, one functional gene may activate a large number of downstream genes in neighboring cells. Sparsifying the number of interacting features (feature sparsity) has the potential to improve causal discovery in biological systems, which remains to be explored.

- While a large number of methods are designed for causal discovery in time-series data, only a limited number of present works aim for causal discovery in general graph-structured data. Time-series based methods cannot be directly adopted on data with multi-branch trajectory dynamics or spatial structures.

**Our contribution.** In this work, we present GEASS (Granger fEAture Selection of Spatiotemporal data), which identifies causally interacting features of high dimensional temporal / spatial data by a single neural network. GEASS considers the aforementioned feature sparsity instead of edge sparsity, thus selects most significant interacting features for downstream causal discovery. Our contributions are three-folds.

1. Instead of direct causal discovery in data, we formulate the task as two steps of causal feature selection and causal graph identification. We provide a novel solution of causal feature selection problem in general graph-structured data by the use of modified transfer entropy maximization with theoretical guarantees.

2. In order to solve our proposed optimization problem, we design a novel combinatorial stochastic gate layer to select non-overlapping sparse feature sets with a newly designed initialization procedure.

3. We demonstrate the power of our method by benchmarking it on both temporal data and spatial data of multiple settings. Our method gives accurate and robust causal feature identification and reveals novel biology in real datasets.

## 1.1 RELATED WORKS

**Neural Granger causality.** Despite the large body of work based on linear Granger causal discovery, neural Granger causality still remains an active area of research. Various neural network architectures, such as MLP, sequential model, and attention-based architecture (Tank et al., 2021; Nauta et al., 2019; Khanna and Tan, 2019; Sun et al., 2021), have been proposed for nonlinear Granger causality

discovery. A recent work uses the information of proxy variable to learn latent confounder for Granger causality by a dual-decoder neural network (Yin and Barucca, 2022). One recent biology-oriented work extends the definition of Granger causality to DAGs, where the use of a linear graph neural network is proposed to model underlying Granger causality (Wu et al., 2021). Meanwhile, a neural-ODE based approach has been proposed to reformulate the Granger causality problem in terms of local dependence graph identification (Bellot et al., 2021).

**Causal feature selection.** The task of causal feature selection has been considered by multiple groups. Most works in this category uses constraint-based methods to identify each feature's causal relation with all other features, equivalent of identifying the whole causal graph structure, including VARLINGAM, tsFCI, SVAR-FCI, and PCMCI (Hyvärinen et al., 2010; Entner and Hoyer, 2010; Malinsky and Spirtes, 2018; Moneta et al., 2011; Runge et al., 2019a). Meanwhile, seqICP focus on identifying the direct or indirect cause for each feature assuming sufficient interventions in the dataset (Pfister et al., 2019). SyPI tackles the causal feature selection problem without the assumption of causal sufficiency and avoids issues in multi-hypothesis testing by construction of the correct conditional set (Mastakouri et al., 2021). Finally, Guo et al. (2022) considers dual correction of causal feature selection to control both false positive rates and false negative rates.

## 2 MODIFIED TRANSFER ENTROPY (MTE)

In order to tackle the issue that a neural network may overfit each model therefore overestimates the number of causal interactions, we need a prediction-free loss function that directly indicates causal signficance. In this work, we propose a novel function, modified transfer entropy (mTE), based on transfer entropy (Schreiber, 2000) as a metric of causal interaction significance.

Transfer entropy is a information-theoretic measure of cross dependence (Schreiber, 2000). Consider two vectorized time series $\boldsymbol{x}^t$ and $\boldsymbol{y}^t$ for $t \in 1, ..., T$. In a Markovian model, the transfer entropy from $\boldsymbol{x}$ to $\boldsymbol{y}$ at time $t$ is defined as the mutual information between the present value $\boldsymbol{x}^t$ and the future value $\boldsymbol{y}^{t+1}$, conditioning on $\boldsymbol{y}^t$ to eliminate possible autocorrelation: $\mathrm{TE}_t(\boldsymbol{x}, \boldsymbol{y}) = I(\boldsymbol{x}^t; \boldsymbol{y}^{t+1}|\boldsymbol{y}^t)$.

By the use of mutual information, transfer entropy is able to model general nonlinear dependencies beyond linear Granger causality. In this work, we further consider the generalization of transfer entropy on graph structured $\boldsymbol{x}^i$ and $\boldsymbol{y}^i$, where $i$ denotes a vertex on the data graph $G = (V, E)$:

$$\mathrm{TE}_i(\boldsymbol{x}, \boldsymbol{y}) := I(\boldsymbol{x}^i; \boldsymbol{y}^{N(i)}|\boldsymbol{y}^i), \quad \text{where } N(i) := \{j|(i, j) \in E\}. \tag{1}$$

Note here the graph can be either directed (the time-series case) or undirected (the spatial case). In this study, we introduce a novel function, modified transfer entropy, that enables the application of bivariate transfer entropy for causal discovery in high-dimensional data. Our key insight is to consider two feature subsets in the dataset that maximizes the mutual information difference:

**Definition 2.1.** *Let $X = [\boldsymbol{x}^1 \boldsymbol{x}^2 \ldots \boldsymbol{x}^n] \in \mathbb{R}^{p \times n}$ be a matrix containing graph-structured vector series $\boldsymbol{x}_i$, with $i$ as vertices of the data graph $G = (V, E)$. Suppose $S_1$ and $S_2$ be two subsets of $\{1, 2, ..., p\}$. The modified transfer entropy $\mathrm{mTE}_i(S_1, S_2)$ and its maximum $\mathrm{mTE}_i^*$ are defined by*

$$\mathrm{mTE}_i(S_1, S_2) := I(\boldsymbol{x}_{S_1}^i; \boldsymbol{x}_{S_2}^{N(i)}) - I(\boldsymbol{x}_{S_1}^i; \boldsymbol{x}_{S_2}^i); \quad \mathrm{mTE}_i^* := \max_{S_1, S_2} \mathrm{mTE}_i(S_1, S_2). \tag{2}$$

Note the mTE function requires strictly stronger dependence than the analogically defined transfer entropy $\mathrm{TE}_i(S_1, S_2)$, as shown by the proposition below (The proof can be seen at Appendix A.1):

**Proposition 2.2.** $\forall S_1, S_2 \subset \{1, ..., p\}, \mathrm{mTE}_i(S_1, S_2) > 0 \Rightarrow \mathrm{TE}_i(S_1, S_2) > 0$.

Let $(S_1^*, S_2^*)$ be one of the maximizers with the smallest size of $|S_1 \cup S_2|$, and denote $S^* := S_1^* \cup S_2^*$ (note $(S_1^*, S_2^*)$ may not be unique). Under some mild assumptions listed below, we are able to provide the theoretical justification for mTE maximization in the time-series setting (Theorem 2.4). A proof can be seen in Appendix A.3.

**Assumptions:**

A1-A3 **Causal Markov assumption, faithfulness, and causal sufficiency** for the causal graph.

A4 **Ergodicity and Stationarity** of the stochastic process defined by the causal graph, meaning the ensemble average equals time average, and the functional relationships encoded by the causal graph do not change by time (or location). This also leads to $\mathrm{mTE}_i(S_1, S_2)$ is constant across $i$.

A5 **DAG causal graph:** We assume $X^T = [t_1, ..., t_m, u_{m+1}, ..., u_p]$ up to a permutation, where $t_i$ are causally interacting features forming a directed acyclic graph (DAG), and $u_k$ are nuisance features that may correlate with $t_i$. An illustration based on the time series setting can be seen in Figure 1.

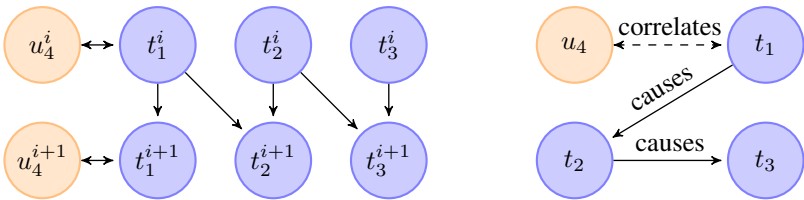

Figure 1: Dependence graph for a single forward step (left) and the underlying causal graph (right).

A6 **Interaction regularity:** Given two disjoint feature sets $A, B$, such that $A$ is a subset of the parent features of $B$ or $B$ is a subset of child features of $A$. Then conditioning on any other feature set $C$ such that $I(A^i, B^{N(i)}|C^i), I(A^i, B^{N(i)}|C^{N(i)}) > 0$, we have:

$$\forall i, \min\{I(A^i, B^{N(i)}|C^i), I(A^i, B^{N(i)}|C^{N(i)})\} > I(A^i, B^i|C^i). \tag{3}$$

**Remark 2.3.** Here our only additional assumption from prevalent literatures (Pearl, 2009; Spirtes et al., 2000) is A6, which aims to filter out features with spurious causations and regularize the algorithmic complexity of causal interactions, thus enabling information-theoretic analysis. A6 has direct connections with the concept of conditional transfer entropy (Faes et al., 2016; Shahsavari Baboukani et al., 2020); further discussions can be seen at Appendix A.2.

**Theorem 2.4.** *Given A1-A6, $S^* := (S_1^* \cup S_2^*) \subseteq \{1, ..., m\}$ (the index set of true interacting features described in A5). Moreover, each feature in $S^*$ is connected to other features in the set $S^*$.*

## 3 NEURAL OPTIMIZATION OF MODIFIED TRANSFER ENTROPY

With Theorem 3.1 stated below, we are able to give a theoretical guarantee of the $l_0$-penalized optimization of mTE. A proof can be seen at Appendix A.4. Here $\odot$ stands for the Hardmard product.

**Theorem 3.1.** *Assume A1-A6 holds and $f, g, h$ define one-to-one mappings on $X \odot \mathbb{1}_{S_1}$ (for $f$) or $X \odot \mathbb{1}_{S_2}$ (for $g, h$). Then $\exists \lambda > 0$, such that for (4), any solution $(S_1^* \cup S_2^*)$ satisfies $S^* := (S_1^* \cup S_2^*) \subseteq \{1, ..., m\}$. Moreover, each feature in $S^*$ is connected to other features in the set.*

$$\min_{f,g,h,S_1,S_2} -(I(f(\boldsymbol{x}^i \odot \mathbb{1}_{S_1}); h(\boldsymbol{x}^{N(i)} \odot \mathbb{1}_{S_2})) - I(f(\boldsymbol{x}^i \odot \mathbb{1}_{S_1}); g(\boldsymbol{x}^i \odot \mathbb{1}_{S_2}))) + \lambda|S_1 \cup S_2| \tag{4}$$

**Remark 3.2.** The estimation of mutual information by various approaches is an active field itself (Belghazi et al., 2018; Hjelm et al., 2018; McAllester and Stratos, 2020; Zhang et al., 2019). In contrast, by this theorem, we show that an accurate estimation of the transfer entropy (such as in (Zhang et al., 2019)) may not be needed as optimizing the **upper bound** of the modified transfer entropy automatically gives the best feature subset selection.

**Remark 3.3.** Our theoretical guarantee is derived based on one-to-one embeddings $f, g, h$. In a neural network, the injectivity may be enforced with various architecture designs yet may not perfectly hold. Empirically, we have found that the optimization of mTE is robust to the embedding injectivity, compared with the original transfer entropy. This is due to our stricter design of the mTE function (Proposition 2.2) and is further illustrated by our experiments in the next section.

Given Theorem 3.1, we are able to construct a neural network for optimizing the proposed loss function. However, the estimation of mutual information is not directly tractable. In this case, because mutual information is invariant by one-to-one transforms, we can restrict the function class of $f, g, h$ in the optimization problem (4) as flows transforming the original feature distributions into Gaussian distributions with fixed dimensionality. We are able to formulate the target for neural network optimization by the explicit formula for mutual information between Gaussians: $I(X, Y) = \frac{1}{2} \log \frac{\det \Sigma_X \det \Sigma_Y}{\det \Sigma_{[X,Y]}}$. The Gaussian regularization can be applied either by regularizing over the discrepancy between embedding distributions $[f, g, h]$ and Gaussian distributions or by applying a adversarial training procedure. In this work, we have implemented the former approach, constructing means and covariance matrices for the concatenated embedding as learnable parameters and minimize the cross entropy between target distributions and the parametrized Gaussian distributions.

## 3.1 COMBINATORIAL STOCHASTIC GATES

In order to solve the optimization problem, we need to learn two sparse sets $S_1, S_2$, which involves combinatorial optimization, making the task impractical for high-dimensional data. To overcome this issue, we use a stochastic gate based approach (Yamada et al., 2020; Lindenbaum et al., 2021), which performs probabilistic relaxation of deterministic $l_0$ norms. In order to explicitly construct $S_1$ and $S_2$ by stochastic gates, we define two random vectors $T^1$ and $T^2$ ranging in $[0, 1]$ with lengths equal to the feature number, with each element independently sampled from STG distribution defined as: $T^i_d = \max(0, \min(1, \mu^i_d + \epsilon^i_d))$, where $\epsilon^i_d \sim N(0, \sigma^2_i)$ is i.i.d. sampled with fixed variance and $\mu^i_d$ is a parameter trainable by reparametrization (Miller et al., 2017; Figurnov et al., 2018).

The new loss function applying stochastic gates can be formulated as:

$$\mathbb{E}_{T^1, T^2} - [\hat{I}(f(\tilde{X}_{S_1}); h(W\tilde{X}_{S_2})) - \hat{I}(f(\tilde{X}_{S_1}); g(\tilde{X}_{S_2}))] + \sum_{d=1}^{p}[\lambda_1 \mathbb{P}(T^1_d > 0) + \lambda_2 \mathbb{P}(T^2_d \in (0, 1))],$$

$$s.t. \quad \tilde{X}_{S_1} = X \odot T^1 \odot T^2, \quad \tilde{X}_{S_2} = X \odot T^1 \odot (1 - T^2). \quad (5)$$

Here $\hat{I}$ is defined as the empirical Gaussian mutual information: $\hat{I}(X, Y) = \frac{1}{2} \log \frac{\det \hat{\Sigma}_X \det \hat{\Sigma}_Y}{\det \hat{\Sigma}_{[X,Y]}}$, and $W$ is defined as the graph diffusion operator: $W\boldsymbol{x}^i = \boldsymbol{x}^{N(i)}$. In our construction, $T^1$ controls the sparsity of feature selection, while $T^2$ controls the expectation of overlap between $\tilde{X}_{S_1}$ and $\tilde{X}_{S_2}$. Denoting the Gaussian error function as $\mathrm{erf}()$, the regularization term for the first layer is of form:

$$\sum_{d=1}^{p} \mathbb{P}(T^1_d > 0) = \sum_{i=1}^{p}(\frac{1}{2} - \frac{1}{2}\mathrm{erf}(\frac{\mu^1_d}{\sqrt{2}\sigma_1})). \quad (6)$$

The regularization term for the second layer can be expressed as:

$$\sum_{d=1}^{p} \mathbb{P}(T^2_d \in (0, 1)) = \sum_{d=1}^{p} \mathbb{P}(T^2_d > 0) - \mathbb{P}(T^2_d \geq 1) = \frac{1}{2} \sum_{d=1}^{p}(\mathrm{erf}(\frac{\mu^2_d}{\sqrt{2}\sigma_2}) - \mathrm{erf}(\frac{\mu^2_d - 1}{\sqrt{2}\sigma_2})). \quad (7)$$

We are able to show strong consistency for our stochastic-gate based feature selection scheme by the theorem below (A proof can be seen at Appendix A.5):

**Theorem 3.4.** *Assume A1-A6 and $f, g, h$ are one-to-one Gaussian embeddings as described above. For the optimal solution of (5), denote a sample of stochastic gate as $T^1, T^2$ and denote the ground truth interacting feature set as $S$, then there exists $\lambda_1, \lambda_2 > 0$ for (5) such that as $n \to \infty$,*

$$\forall i \in \{0, 1\}, \ \mathbb{P}(B_i \subseteq S) \xrightarrow{a.s.} 1, \text{ where } B_i := \{d | T^1_d > 0, T^2_d = i\}. \quad (8)$$

In practice, we also have observed the method's solution highly depends on the stochastic gate initialization. Here we provide a heuristic initialization scheme that shows superior empirical performance. Details of the initialization scheme can be seen in Appendix B.

## 3.2 PROPOSED NETWORK ARCHITECTURE

Our proposed network architecture is summarized in Figure 2. For an input dataset $X \in \mathbb{R}^{p \times n}$ and its corresponding graph adjacency matrix $A \in \mathbb{R}^{n \times n}$, we first pass each feature through two sequential stochastic gate layers $T^1, T^2$. The $l_0$ penalty is conducted on the first STG layer, while the second STG layer is regularized with the 0-1 penalty, consistent with the descriptions in the previous section.

After passing each feature, denote $\hat{T}^2_i = 1 - T^2_i$, we have two intermediate embeddings defined by $\tilde{X}_{S_1} = X \odot T^1 \odot T^2$ and $\tilde{X}_{S_2} = X \odot T^1 \odot \hat{T}^2$ respectively. Then these two embeddings are passed through MLP1 ($f$) and MLP2 ($g$) to generate Gaussian embeddings $f(\tilde{X}_{S_1}), g(\tilde{X}_{S_2})$ corresponding to (5). For the design of function $h$, we consider two crucial elements: 1. an additional layer to aggregate the information from different nodes in $\boldsymbol{x}^{N(i)}$; 2. the injectivity of mappings $f, g, h$. Note $f, h$ in (5) are automatically enforced to be injective on interacting features to maximize the first term of mTE, but $g$ is not. Therefore, our final design of $h$ is the composition of first applying $g$ (enforcing the injectivity of $g$), a mean aggregation layer without self-loop consistent with the GCN design (Kipf and Welling, 2016) by multiplying the adjacency matrix $A$, and another MLP layer (MLP3). Finally, we compute the minus empirical Gaussian mTE $\hat{I}(f, g) - \hat{I}(f, h)$ and add the cross-entropy penalty between the concatenated embedding distribution and a learnable Gaussian distribution.

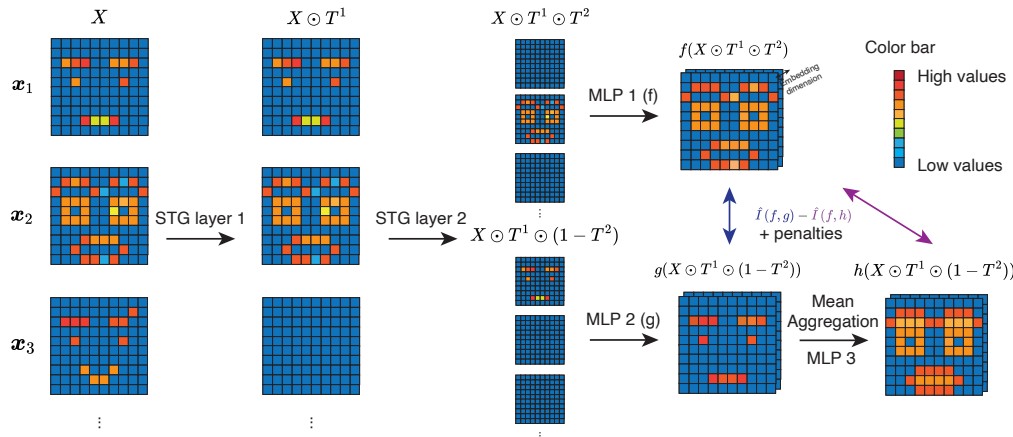

Figure 2: Illustration of the proposed GEASS network architecture.

### 3.3 OUTPUT INTERPRETATION

Upon the algorithm convergence, GEASS provides both outputs of active features ($B_0 \cup B_1$) and embeddings ($f, g, h$) produced by causally interacting features. In this paper, we emphasize the use of the identified interacting features $B_0 \cup B_1$. The output of embeddings ($f, g, h$) may be complex and nonlinear, potentially requiring additional architectures to maximize its interpretability.

By the construction of GEASS, we are able to get two separate sparse feature subsets as source features $B_1$ and sink features $B_0$. These features may be used as inputs to further proper causal analysis, such as LPCMCI (Gerhardus and Runge, 2020) for time-series data, which despite its statistical power in depicting possible lags, identifying latent confounders, and allowing nonlinear tests, can only work on data with moderate feature sizes. Also, these features may be used in other machine learning models for improved model interpretability.

## 4 EXPERIMENTS

### 4.1 GAUSSIAN TIME-SERIES WITH POSSIBLE NONLINEARITY

In order to benchmark the method in time-series data, we consider two settings: 1. Minor effect of latent processes, with autocorrelation present; 2. Significant effect of latent processes, with autocorrelation present. Both settings are modeled by Gaussian structural processes with an underlying causal graph. Further details can be seen in Appendix C.1.

We test the false discovery rate (FDR) and F1 score between ground truth interacting features and recovered features as two metrics for high-dimensional data causal discovery. We compare GEASS with two categories of methods, namely conditional independence based (CI-based) methods and Granger causality based (GC-based) methods respectively. The first method category includes VAR-LINGAM (Hyvärinen et al., 2010), PCMCI (Runge et al., 2019b), and LPCMCI (Gerhardus and Runge, 2020). Among them, despite the statistical power, LPCMCI is not included in our experiment as it fails to converge in given time in our preliminary experiments. The second method category includes a neural-network based generalized vector autoregression model GVAR Granger (Marcinkevičs and Vogt, 2021), and Grid-net which generalizes the definition of Granger causality to Directed Acyclic Graph (DAG) (Wu et al., 2021); moreover we include two state-of-the-art approaches, DCM and NGM implemented in (Bellot et al., 2021) that use neural ODE to model nonlinear dependence graph.

Table 1 shows our benchmarking results. Among the alternative methods, GVAR and GrID-net fail in all settings as they are not designed for causal feature selection. VAR-LINGAM achieves high accuracy in linear settings while fails in nonlinear settings. In contrast, PCMCI fails when latent processes contribute to both true causally interacting features and nuisance features, creating spurious correlations. Empirically we also observe that DCM and NGM achieves comparable performance

when the dynamics are linear but performs worse in the nonlinear setting, where the dynamics are more irregular. Finally, GEASS consistently gives accurate causal feature identifications (high F1) and low false discovery rate (low FDR) in all settings considered.

Table 1: Comparison of methods on Gaussian linear / nonlinear time-series data with different feature numbers and different nuisance feature settings.

| | Weak confounding interactions | | | | Strong confounding interactions | | | |
| | Linear | | nonlinear | | Linear | | nonlinear | |
| Methods | FDR | F1 score | FDR | F1 score | FDR | F1 score | FDR | F1 score |
|---|---|---|---|---|---|---|---|---|
| LINGAM (CI) | .00 (.00) | .94 (.04) | .83 (.00) | .17 (.00) | .00 (.00) | .94 (.04) | .83 (.00) | .17 (.00) |
| PCMCI (CI) | .17 (.01) | .81 (.04) | .12 (.08) | .85 (.05) | 1.0 (.00) | .00 (.00) | .63 (.23) | .36 (.22) |
| GVAR (GC) | .94 (.00) | .11 (.00) | .94 (.00) | .11 (.00) | .94 (.00) | .11 (.00) | .94 (.00) | .11 (.00) |
| GrID-net (GC) | 1.0 (.00) | .00 (.00) | 1.0 (.00) | .00 (.00) | 1.0 (.00) | .00 (.00) | 1.0 (.00) | .00 (.00) |
| DCM (GC) | .12 (.20) | .88 (.20) | .65 (.12) | .35 (.12) | .18 (.09) | .82 (.09) | .93 (.11) | .07 (.11) |
| NGM (GC) | .07 (.08) | .88 (.04) | .48 (.17) | .50 (.17) | .00 (.00) | .91 (.00) | .62 (.25) | .38 (.25) |
| **GEASS (Ours)** | .05 (.15) | .97 (.10) | .03 (.06) | .92 (.05) | .03 (.07) | .90 (.04) | .00 (.00) | .91 (.00) |

Furthermore, we evaluate different methods' scalability with respect to the feature size. (Experimental details can be seen at Appendix C.1.2). As described before, we anticipate high computational complexity of both conditional independence based methods and neural network based methods with respect to the feature size, which prohibits further use of these methods for high-dimensional biological data analysis, where the feature number is typically at the scale of $10^3 - 10^4$. Meanwhile, GEASS constructs a single neural network with parameters approximately proportional to $p$, thus largely reducing the complexity in the high-dimensional regime. We benchmark PCMCI, GVAR, GrID-net, NGM, GEASS, and an additional combination of GEASS with a downstream CI-test based causal graph identification method LPCMCI. Our experimental result shows the superior performance of GEASS as well as GEASS+LPCMCI in time complexity, consistent with our qualitative analysis (Figure 3).

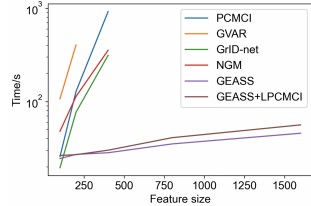

Figure 3: Running time comparison for methods on time-series data with different feature sizes.

### 4.2 SIMULATED SPATIAL OMICS DATA WITH CELL TYPE CONFOUNDER

In order to jointly consider spatial confounders and corresponding autocorrelation patterns that are potentially enriched in specific niches, we consider the case of spatial omics data, where the autocorrelation is modeled by a higher likelihood of same type of cells in the neighborhood, and the confounder (nuisance features) is modeled by a coherent shift of global gene expression for each cell type. We first simulate scRNA-seq datasets, then each synthetic scRNA-seq dataset is assigned to a fixed size grid with cell type labels simulated by Ising model simulation. We then add artificial genes that are spatially correlated with neighboring cell's given gene set. Finally each dataset is normalized and log1p transformed as the standard pipeline in Scanpy (Wolf et al., 2018).

The majority of methods are not available as their focus is on time-series data. Therefore in order to perform our benchmarking study, we compare GEASS with Lasso Granger, as well as our implemented L1-regularized version of NCEM, an approach proposed to detect interactions in spatial omics data (Fischer et al., 2021). Finally, we also implemented a method that maximizes over the original transfer entropy to select causal features (TE).

As shown in Table 2, the original LASSO cannot identify causal features because of the strong correlation between features. L1-NCEM alleviates the issue by conditioning on cell type labels in regression. TE outperforms linear methods yet generates a number of false positives, as it may learn spurious causations as discussed in Remark 3.3. Finally, GEASS consistently outperforms over other methods in identifying causal features of data as shown by both high F1 score and low FDR.

Table 2: Comparison of methods on simulated spatial transcriptomics data.

| | Linear | | nonlinear | |
|---|---|---|---|---|
| Methods | FDR | F1 score | FDR | F1 score |
| Lasso | 0.950±0.055 | 0.050±0.055 | 0.970±0.040 | 0.030±0.040 |
| L1-NCEM | 0.380±0.138 | 0.620±0.138 | 0.535±0.134 | 0.465±0.134 |
| TE | 0.190±0.127 | 0.767±0.070 | 0.141±0.087 | 0.761±0.060 |
| GEASS (Ours) | **0.095±0.128** | **0.787±0.088** | **0.000±0.000** | **0.775±0.110** |

### 4.3 SCRNA-SEQ PANCREATIC ENDOCRINOGENESIS TRAJECTORY

We test GEASS on the pancreatic endocrinogenesis trajectory data, which is a standard dataset for scRNA-seq trajectory inference task (Bergen et al., 2020; Bastidas-Ponce et al., 2019). The pancreas trajectory data contains 3696 cells and 27998 genes. After preprocessing, lowly-expressed genes are filtered out as the standard pipeline in scVelo (Bergen et al., 2020), with remaining 2000 genes for further analysis. We aim to use GEASS to identify causally-related genes along the developmental trajectory to reveal underlying biology. (See Appendix C.3 for experimental details).

scRNA-seq data provides a snapshot of cell population distribution therefore time-series based analysis methods cannot be directly applied. However, due to GEASS's flexible setting in forward operator $W$, we are able to define the time flow by RNA velocity analysis. RNA velocity analysis uses the additional information of intron RNAs to infer the underlying dynamics of gene expression change. Thus, we are able to define a velocity kernel matrix $A_{velo}$, which provides weighted adjacency relationships of cells based on velocity direction and cell phenotypic proximity.

GEASS identifies 50 causally-related features with high biological relevance. For example, the gene list includes the key transcriptional regulator *NEUROG3*, which is required for the specification of a common precursor of the 4 pancreatic terminal states (uni, 2021). As the ground truth causal interactions here are unknown, for further quantitative validation, we assume the underlying biological process is driven by a causal cascade of gene interactions, meaning target genes activated in earlier phases of the trajectory further cause downstream gene activation at later phases. In this case, the higher a gene velocity is, the more likely the gene is associated with causal gene-gene relationships. Our benchmarking result here suggests GEASS achieves the best performance in selecting genes with high mean velocity likelihood, compared with alternative gene selection schemes with fixed gene number (50) including high-expressed genes (HEG), highly-variable genes (HVG), and genes having high correlation with inferred latent time (HCG) (Figure 4).

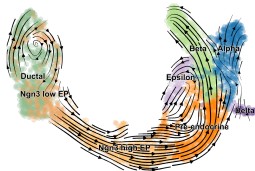

| | Mean RNA velocity likelihood |
|---|---|
| HEG | 0.0528 |
| HVG | 0.1753 |
| HCG | 0.1889 |
| GEASS | **0.2366** |

Figure 4: Visualization of pancreas trajectory dataset and comparisons of gene selection criterions.

### 4.4 MERFISH HUMAN CORTEX SINGLE-CELL LEVEL SPATIAL TRANSCRIPTOMICS

Spatial transcriptomics represent a wide category of method that can achieve spatial profiling of gene expression in tissues (Moses and Pachter, 2022; Rao et al., 2021; Palla et al., 2022). By the additional information of spatial locations, such measurements enable deeper understandings of cellular interactions (Palla et al., 2022; Jerby-Arnon and Regev, 2022; Fischer et al., 2021). However, current computational methods revealing interaction modules (Jerby-Arnon and Regev, 2022) or niche effects (Fischer et al., 2021; Raredon et al., 2023) for spatial omics data lacks causal interpretation. Applying GEASS, we aim to reveal underlying causal intercellular patterns to fully utilize the potential of spatial omics data for biological discovery.

Here we use GEASS on a recent published MERFISH dataset measuring spatially-resolved single-cell gene expression of human cortex (Fang et al., 2022). The dataset we used comprises of 3044 cells and 4000 genes; each cell is annotated as one of the eight cell types: excitatory neurons (EXC), inhibitory neurons (INC), astrocytes (ASC), microglial cells (MGC), oligodendrocytes (OGC), oligodendrocyte progenitor cells (OPC), endothelial cells (ENDO), and mural cells (MURAL) as shown by the first panel of Figure 6 in Appendix D. Our GEASS analysis selects 9 genes, namely *FILIP1, SLC17A7, MYH11, RP11-10j21.2, PIRT, C3ORF67, TRDMT1, RGS8, SPTLC2* (Appendix Figure 6), with further experimental details available in Appendix C.4. Among these genes, *MYH11*, *RP11-10j21.2*, and *TRDMT1* are enriched at the endothelial cells adjacent with mural cells, corresponding to underlying vascular structures (marked by ellipses in the first panel of Appendix Figure 6). We next aim to verify if their expression difference with those of non-adjacent endothelial cells is statistically significant. Indeed, by applying the Wilcoxon rank-sum test, we have found significant enrichments for both *MYH11* and *TRDMT1*, with p-values 0.003 and 0.015 respectively, while the p-value for the gene *RP11-10j21.2* is not significant (0.5) due to the gene expression sparsity. The finding is consistent with the MERFISH images, which reveals rich cellular interactions between neuronal cells and the blood vessels (Fang et al., 2022). Therefore, these identified marker genes of vascular structure may encode underlying meaningful cellular interactions.

Next, we focus on two GEASS identified genes, *C3ORF67* and *PIRT*, which are highly expressed at nearby spatial locations. In order to confirm the possible causal relationship between the two genes, we consider three models: 1. the two genes are expressed in the same cell without spatial causal relationships; 2. The expression of *C3ORF67* in each cell causes the expression of *PIRT* in neighboring cells (*C3ORF67* → *PIRT*); 3. The expression of *PIRT* in each cell causes the expression of *C3ORF67* in neighboring cells (*PIRT* → *C3ORF67*). To this end, we first compare Pearson and Spearman p-values of intracellular correlation (model 1), *C3ORF67* to neighboring *PIRT* (model 2), and *PIRT* to neighboring *C3ORF67* (model 3). Our comparison shows for the p-values of

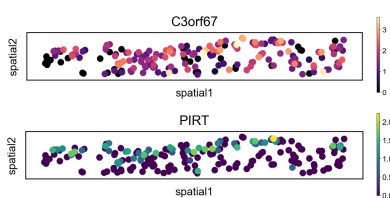

Figure 5: Normalized spatial expression levels of genes *C3orf67* and *PIRT*.

both correlation measures, model 3 is favored (0.004, 0.001) over model 1 (0.014, 0.003) and model 2 (0.049, 0.004). The validity of model 3 (*PIRT* → *C3ORF67*) is further supported by a linear model predicting *C3ORF67* expression by both intracellular and neighbor expression of *PIRT*, where the neighboring cell effect coefficient is significant at the confidence level of 0.01 by bootstrap, while the alternative model's corresponding coefficient is not significant. Our finding is consistent with the predicted role of *PIRT* in transmembrane transporter binding and phosphatidylinositol-mediated signaling (Safran et al., 2021). As the role of *C3ORF67* in human cortex remains unclear, this revealed causal link may lead to novel biological discoveries with further experimental validations.

## 5 CONCLUSIONS

In this work, we present GEASS, a causal feature selection method based on information-theoretic tools and neural networks. GEASS is able to scale to high dimensions and identify sparse interacting features. We provide both theoretical gaurantees and empirical validations of GEASS on synthetic and real biological data. Our results show GEASS can be integrated into high-dimensional spatiotemporal data analysis pipelines to provide unique insights for further findings.

**Limitations.** GEASS is a method designed for nonlinear causal feature selection. GEASS does not provide a causal graph itself as it optimizes a latent embedding corresponding to different causal mechanisms. Therefore, in applications where a causal graph output is favored, constraint-based methods may need to be applied after GEASS. Moreover, when underlying causal graph has a large number of vertices, the sparsity assumption is violated and GEASS is not gauranteed to work. Also, further efforts may be taken to incorporate lag selections for GEASS.

**Broader impact.** We anticipate a wide use of GEASS in high-dimensional graph-structured data, especially for high-dimensional biological data such as single cell trajectories and spatial omics measurements. Applying GEASS along with causal graph identification methods to a wider range of real biological data may greatly facilitate downstream biological discoveries.

ACKNOWLEDGEMENTS

The authors thank Ofir Lindenbaum, Boaz Nadler, Yifei Min, and Ronen Basri for helpful discussions. Y.K. acknowledges support by NIH grants R01GM131642, UM1DA051410, U54AG076043, P50CA121974, and U01DA053628.

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

## APPENDIX

## A   PROOFS

### A.1   PROOF OF PROPOSITION 2.2.

**Proposition 2.2**. $\forall S_1, S_2 \subset \{1, ..., p\}, \mathrm{mTE}_i(S_1, S_2) > 0 \Rightarrow \mathrm{TE}_i(S_1, S_2) > 0$.

*Proof.* By standard properties of mutual information (Cover, 1999) we have

$$
\begin{aligned}
\mathrm{TE}_i(X_{S_1}, X_{S_2}) &= I(X_{S_1}^i; X_{S_2}^{j:(i,j)\in E} | X_{S_2}^i) \\
&= I(X_{S_1}^i; X_{S_2}^{j:(i,j)\in E}, X_{S_2}^i) - I(X_{S_1}^i; X_{S_2}^i) \\
&= I(X_{S_1}^i; X_{S_2}^{j:(i,j)\in E}) - I(X_{S_1}^i; X_{S_2}^i) + I(X_{S_1}^i; X_{S_2}^i | X_{S_2}^{j:(i,j)\in E}).
\end{aligned}
\tag{9}
$$

Therefore $\mathrm{TE}_i(S_1, S_2) \geq \mathrm{mTE}_i(S_1, S_2)$ holds, thus $\mathrm{mTE}_i(S_1, S_2) > 0 \Rightarrow \mathrm{TE}_i(S_1, S_2) > 0$.   $\square$

### A.2   DISCUSSION OF ASSUMPTION A6.

Our assumption A6 is based on the concept of conditional mutual entropy, which aims to filter out possible indirect causal relationships.

Here are two simple examples to see why TE/mTE can have problems with indirect causal interactions in the time-series setting: consider the relationships: $s_t \to w_t \to v_{t+1}; s_t \to w_{t+1} \to v_{t+1}$. Then in both cases, we may have: $I(s_t, v_{t+1}) - I(s_t, v_t) > 0$ and $I(s_t, v_{t+1}|v_t) > 0$ although there are no direct causal relationship between $s$ and $v$. Note in our setting, we include the possibility of such indirect interaction by allowing correlation between nuisance features and true interacting features.

The issue can be resolved by considering the conditional mutual information $I(s_t, v_{t+1}|w_t)$ or $I(s_t, v_{t+1}|w_{t+1})$, which equals 0. This insight is also addressed the concept of conditional transfer entropy:

**Definition (Conditional transfer entropy)** (Shahsavari Baboukani et al., 2020). Assume $X$ and $Y$ are the features of interest and the conditioning features are $Z$. Denote $-$ as $[1, 2, ..., t]$, then we have

$$
\mathrm{cTE}_t(X, Y, Z) = I(Y_{t+1}, X_- | Y_-, Z_-).
$$

The classical formulation of conditional transfer entropy is widely used in high-dimensional observational data to learn direct causal dependencies (Faes et al., 2016; Shahsavari Baboukani et al., 2020). It implicitly assumes that, there is direct causal relationship between $X$ and $Y$ if $\forall Z, t, \mathrm{cTE}_t(X, Y, Z) > 0$. Here, we extend this assumption in the context of conditional mTE covering both examples described above. The conditional mTEs are defined in analogy to cTE for generalized graph-structured data in the Markovian model setting:

**Definition (Two forms of conditional mTE)**. Assume $X$ and $Y$ are the feature sets of interest and the conditioning features are $Z$. Then we have

$$
\mathrm{cmTE}_i^1(X, Y, Z) = I(X^i, Y^{N(i)} | Z^i) - I(X^i, Y^i | Z^i);
$$

$$\text{cmTE}_i^2(X, Y, Z) = I(X^i, Y^{N(i)}|Z^{N(i)}) - I(X^i, Y^i|Z^i);$$

By controlling the two forms of conditional mTE to be larger than zero, we rule out both possibilities of $X^i \to Z^i \to Y^{N(i)}$ and $X^i \to Z^{N(i)} \to Y^{N(i)}$, as mTE is a stricter version of the original transfer entropy as discussed in Proposition 2.2. In summary, our A6 can be reformulated as $\forall Z, i, \text{cmTE}_i^1(X, Y, Z) > 0; \text{cmTE}_i^2(X, Y, Z) > 0$ for ground truth interacting $X, Y$ in non-degenerating cases, where $Z$ does not fully overlap with $X/Y$ in the same point.

### A.3  PROOF OF THEOREM 2.4.

**Theorem 2.4**. *Given A1-A6, $S^* := (S_1^* \cup S_2^*) \subseteq \{1, ..., m\}$ (the index set of true interacting features described in A5). Moreover, each feature in $S^*$ is connected to other features in the set $S^*$.*

*Proof.* **Step 1.** First we prove $S_1^* \cap S_2^* = \emptyset$. If not, assume $p$ is an overlapping element. For simplicity, we denote $N(i) := \{j|(i, j) \in E\}, A = X_{S_1^*}, B = X_{S_2^*}$. Then we have

$$
\begin{aligned}
&\text{mTE}(S_1^*, S_2^*) - \text{mTE}(S_1^* \setminus p, S_2^*) \\
&= I(A^i \setminus p^i, p^i; B^{N(i)} \setminus p^{N(i)}, p^{\{N(i)\}}) - I(A^i, p^i; B^i, p^i) - I(A^i \setminus p^i; B^{N(i)} \setminus p^{N(i)}, p^{N(i)}) \\
&\quad + I(A^i; B^i, p^i) \\
&= I(p^i; B^{N(i)} \setminus p^{N(i)}, p^{N(i)}|A^i \setminus p^i) - I(p^i; B^i \setminus p^i, p^i|A^i \setminus p^i) < 0.
\end{aligned}
\tag{10}
$$

Therefore removing $p$ would increase the value of mTE, leading to a contradiction.

**Step 2.** Now we prove nuisance signals cannot be in either $S_1^*$ or $S_2^*$. Otherwise, first we assume a set of nuisance signals $U$ is in $S_1^*$. Here we denote $A := X_{S_1^*}, B := X_{S_2^*}$. As $U$ only interacts with variables at the same time point, $U$ can only interact with $B^{N(i)}$ via indirect links through a subset of interacting features at $i$. Denote this feature set as $Pa_U(B)^i \subseteq \{t_1^i, ..., t_m^i\}$, and the difference set $Pa_U^-(B)^i := Pa_U(B)^i \setminus B^i$. Then we first note $Pa_U^-(B)^i$ cannot be an empty set. Otherwise, denote $S_1 := S_1^* \setminus U$, noting the non-overlapness between $A$ and $B$ we would have

$$
\begin{aligned}
&\text{mTE}(S_1^*, S_2^*) - \text{mTE}(S_1, S_2^*) \\
&= I(A^i \setminus U^i, U^i; B^{N(i)}) - I(A^i \setminus U^i, U^i; B^i) - I(A^i \setminus U^i; B^{N(i)}) \\
&\quad + I(A^i \setminus U^i; B^i) \\
&= I(U^i; B^{N(i)}|A^i \setminus U^i) - I(U^i; B^i|A^i \setminus U^i) \\
&= -h(U^i|B^{N(i)}, A^i \setminus U^i) + h(U^i|B^i, A^i \setminus U^i) \\
&\leq -h(U^i|B^{N(i)}, A^i \setminus U^i) + h(U^i|Pa_U(B)^i, A^i \setminus U^i) \text{ (Conditioning reduces entropy)} \\
&\leq 0.
\end{aligned}
\tag{11}
$$

This means $(S_1, S_2^*)$'s mTE is not smaller than $(S_1^*, S_2^*)$'s while having a smaller union size, leading to a contradiction. Then because $Pa_U^-(B)$ does not overlap with either $U$ and $B$, with A6 we have

$$
\begin{aligned}
&\text{mTE}(S_1^*, S_2^*) - \text{mTE}(S_1^* \cup \text{Index}(Pa_U^-(B)), S_2^*) \\
&= I(A^i \setminus U^i, U^i; B^{N(i)}) - I(A^i \setminus U^i, U^i; B^i) - I(A^i \setminus U^i, U^i, Pa_U^-(B)^i; B^{N(i)}) \\
&\quad + I(A^i \setminus U^i, U^i, Pa_U^-(B)^i; B^i) \\
&= I(Pa_U^-(B)^i; B^i|A^i) - I(Pa_U^-(B)^i; B^{N(i)}|A^i) \overset{\text{A6}}{\leq} 0.
\end{aligned}
\tag{12}
$$

The equal sign above is taken iff. $Pa_U^-(B)^i \subseteq A^i$. Further we have

$$
\begin{aligned}
&\mathrm{mTE}(S_1^* \cup \mathrm{Index}(Pa_U^-(B)), S_2^*) - \mathrm{mTE}(S_1 \cup \mathrm{Index}(Pa_U^-(B)), S_2^*) \\
&= I(A^i \setminus U^i, U^i, Pa_U^-(B)^i; B^{N(i)}) - I(A^i \setminus U^i, U^i, Pa_U^-(B)^i; B^i) \\
&\quad - I(A^i \setminus U^i, Pa_U^-(B)^i; B^{N(i)}) + I(A^i \setminus U^i, Pa_U^-(B)^i; B^i) \\
&= I(U^i; B^{N(i)}|Pa_U^-(B)^i, A^i \setminus U^i) - I(U^i; B^i|Pa_U^-(B)^i, A^i \setminus U^i) \\
&= -h(U^i|B^{N(i)}, Pa_U^-(B)^i, A^i \setminus U^i) + h(U^i|B^i, Pa_U^-(B)^i, A^i \setminus U^i) \\
&\leq -h(U^i|B^{N(i)}, Pa_U^-(B)^i, A^i \setminus U^i) + h(U^i|Pa_U(B)^i, A^i \setminus U^i) \leq 0.
\end{aligned}
\tag{13}
$$

Therefore, in all possible cases, $\mathrm{mTE}(S_1 \cup \mathrm{Index}(Pa_U^-(B)^i), S_2^*)$ is either strictly larger than $\mathrm{mTE}(S_1^*, S_2^*)$ or equal with $\mathrm{mTE}(S_1^*, S_2^*)$ but with smaller union size, leading to a contradiction.

Next, given the result above, we assume a nuisance signal set $U$ is in $S_2^*$, and $S_1^*$ does not include any nuisance features. Then as $U$ only interacts with variables at the same time point, $U^{N(i)}$ can only interact with $S_1^*$ via indirect links through a subset of interacting features at $N(i)$. Denote the whole intermediate feature set for $A$ as $Ch_U(A)^{N(i)} \subseteq \{t_1^{N(i)}, ..., t_m^{N(i)}\}$, and $Ch_U^-(A)^{N(i)} := Ch_U(A)^{N(i)} \setminus A^{N(i)}$. Then same as above, denote $S_2 = S_2^* \setminus U$, if $Ch_U^-(A)$ is an empty set we would have

$$
\begin{aligned}
&\mathrm{mTE}(S_1^*, S_2^*) - \mathrm{mTE}(S_1^*, S_2) \\
&= I(A^i; B^{N(i)} \setminus U^{N(i)}, U^{N(i)}) - I(A^i; B^i \setminus U^i, U^i) - I(A^i; B^{N(i)} \setminus U^{N(i)}) \\
&\quad + I(A^i; B^i \setminus U^i, U^i) \\
&= I(A^i; U^{N(i)}|B^{N(i)} \setminus U^{N(i)}) - I(A^i; U^i|B^i \setminus U^i) \\
&= -h(U^{N(i)}|B^{N(i)} \setminus U^{N(i)}, A^i) + h(U^i|B^i \setminus U^i, A^i) \\
&\leq -h(U^{N(i)}|B^{N(i)} \setminus U^{N(i)}, A^i) + h(U^i|Ch_U(A)^i, B^i \setminus U^i) \leq 0.
\end{aligned}
\tag{14}
$$

Above derivation holds due to stationarity (as $|N(i)| \equiv 1$ in the time series setting). Therefore $Ch_U^-(A)$ cannot be an empty set. Because of the non-overlapness between $Ch_U^-(A)$ and either $A$ or $U$, with A6, we have

$$
\begin{aligned}
&\mathrm{mTE}(S_1^*, S_2^*) - \mathrm{mTE}(S_1^*, S_2 \cup \mathrm{Index}(Ch_U^-(A))) \\
&= I(A^i; B^{N(i)} \setminus U^{N(i)}, U^{N(i)}) - I(A^i; B^i \setminus U^i, U^i) \\
&\quad - I(A^i; B^{N(i)} \setminus U^{N(i)}, U^{N(i)}, Ch_U^-(A)^{N(i)}) + I(A^i; B^i \setminus U^i, U^i, Ch_U^-(A)^i) \\
&= I(A^i; Ch_U^-(A)^i|B^i) - I(A^i; Ch_U^-(A)^{N(i)}|B^{N(i)}) \overset{\mathrm{A6}}{\leq} 0.
\end{aligned}
\tag{15}
$$

The equal sign above is taken iff. $Ch_U^-(A)^i \subseteq B^i$. Further we have

$$
\begin{aligned}
&\mathrm{mTE}(S_1^*, S_2^* \cup \mathrm{Index}(Ch_U^-(A))) - \mathrm{mTE}(S_1^*, S_2 \cup \mathrm{Index}(Ch_U^-(A))) \\
&= I(A^i; B^{N(i)} \setminus U^{N(i)}, U^{N(i)}, Ch_U^-(A)^{N(i)}) - I(A^i; B^i \setminus U^i, U^i, Ch_U^-(A)^i) \\
&\quad - I(A^i; B^{N(i)} \setminus U^{N(i)}, Ch_U^-(A)^{N(i)}) + I(A^i; B^i \setminus U^i, Ch_U^-(A)^i) \\
&= I(A^i; U^{N(i)}|B^{N(i)} \setminus U^{N(i)}, Ch_U^-(A)^{N(i)}) - I(A^i; U^i|B^i \setminus U^i, Ch_U^-(A)^i) \leq 0.
\end{aligned}
\tag{16}
$$

Therefore, in all possible cases, $\mathrm{mTE}(S_1^*, S_2 \cup \mathrm{Index}(Ch_U^-(A)))$ is either strictly larger than $\mathrm{mTE}(S_1^*, S_2^*)$ or equal with $\mathrm{mTE}(S_1^*, S_2^*)$ but with smaller union size, leading to a contradiction.

**Step 3.** Moreover, if there exists a component in $S_1^* \cup S_2^*$ not connected to any other feature components, denote the feature as $q$. Then, in this case with A1-4, the feature $q$ is independent of any other features in $S_1^* \cup S_2^*$. From step 1 it can be deduced that $q$ cannot be in both $S_1^*, S_2^*$. Therefore in this case, we have $\mathrm{mTE}(S_1^* - q, S_2^* - q) = \mathrm{mTE}(S_1^*, S_2^*)$ thus leading to the contradiction of finding an $(S_1, S_2)$ with the same mTE but smaller $|S_1 \cup S_2|$.

$\square$

## A.4 PROOF OF THEOREM 3.1.

**Theorem 3.1.** *Assume A1-A6 holds and $f, g, h$ define one-to-one mappings on $X \odot \mathbb{1}_{S_1}$ (for $f$) or $X \odot \mathbb{1}_{S_2}$ (for $g, h$). Then $\exists \lambda > 0$, such that for (4), any solution $(S_1^* \cup S_2^*)$ satisfies $S^* := (S_1^* \cup S_2^*) \subseteq \{1, ..., m\}$. Moreover, each feature in $S^*$ is connected to other features in the set.*

$$\min_{f,g,h,S_1,S_2} -(I(f(\boldsymbol{x}^i \odot \mathbb{1}_{S_1}); h(\boldsymbol{x}^{N(i)} \odot \mathbb{1}_{S_2})) - I(f(\boldsymbol{x}^i \odot \mathbb{1}_{S_1}); g(\boldsymbol{x}^i \odot \mathbb{1}_{S_2}))) + \lambda|S_1 \cup S_2|$$

*Proof.* With A4 (ergodicity and stationarity), the optimization problem 4 is equivalent to

$$\min_{f,g,h,S_1,S_2} -(I(f(\boldsymbol{x}_{S_1}^i); h(\boldsymbol{x}_{S_2}^{N(i)})) - I(f(\boldsymbol{x}_{S_1}^i); g(\boldsymbol{x}_{S_2}^i))) + \lambda|S_1 \cup S_2|. \tag{17}$$

Given the assumption that $f, g, h$ define injective mappings on $\boldsymbol{x}_{S_1}^i, \boldsymbol{x}_{S_2}^i$ respectively, and one-to-one transformation does not change mutual information, we have the optimization problem is equivalent to

$$\min_{S_1,S_2} -(I(\boldsymbol{x}_{S_1}^i; \boldsymbol{x}_{S_2}^{N(i)}) - I(\boldsymbol{x}_{S_1}^i; \boldsymbol{x}_{S_2}^i)) + \lambda|S_1 \cup S_2|. \tag{18}$$

Using Theorem 2.4, a minimizer of the mTE term with the smallest union size satisfies $S^* := (S_1^* \cup S_2^*) \subseteq \{1, ..., m\}$. Moreover, each feature in $S_1^* \cup S_2^*$ is connected to other features in the set. Note that with our definition of optimal $S_1, S_2$, the minimal gap between $\text{mTE}(S_1^*, S_2^*)$ and any other value $\text{mTE}(S_1, S_2)$ with smaller $|S_1 \cup S_2|$ size is larger than zero. Denote the minimal gap as $\delta$, and take $\lambda < \frac{\delta}{|S_1^* \cup S_2^*|}$, then for these other solutions, we have

$$
\begin{aligned}
&- \text{mTE}(S_1, S_2) + \lambda|S_1 \cup S_2| \\
&\geq -\text{mTE}(S_1^*, S_2^*) + \delta + \lambda|S_1 \cup S_2| \\
&\geq -\text{mTE}(S_1^*, S_2^*) + \delta \\
&> -\text{mTE}(S_1^*, S_2^*) + \lambda|S_1^* \cup S_2^*|.
\end{aligned}
\tag{19}
$$

Meanwhile, for the $(S_1, S_2)$ with larger union size, with the definition of the mTE, we have

$$
\begin{aligned}
&- \text{mTE}(S_1, S_2) + \lambda|S_1 \cup S_2| \\
&\geq -\text{mTE}(S_1^*, S_2^*) + \lambda|S_1 \cup S_2| \\
&= -\text{mTE}(S_1^*, S_2^*) + \lambda(|S_1 \cup S_2| - |S_1^* \cup S_2^*|) + \lambda|S_1^* \cup S_2^*| \\
&> -\text{mTE}(S_1^*, S_2^*) + \lambda|S_1^* \cup S_2^*|.
\end{aligned}
\tag{20}
$$

Therefore, when taking $\lambda \in (0, \frac{\delta}{|S_1^* \cup S_2^*|})$, the desired optimal $S_1, S_2$ by mTE is the optimal output of the constructed optimization problem. $\qquad\square$

## A.5 PROOF OF THEOREM 3.4.

**Theorem 3.4.** *Assume A1-A6 and $f, g, h$ are one-to-one Gaussian embeddings as described above. Denote for the optimal solution of (5), a sample of stochastic gate is given by $T^1, T^2$ and denote the ground truth interacting feature set as $S$, then there exists $\lambda_1, \lambda_2 > 0$ for (5) such that as $n \to \infty$,*

$$\forall i \in \{0, 1\}, \ \mathbb{P}(B_i \subseteq S) \xrightarrow{a.s.} 1, \ \text{where } B_i := \{d | T_d^1 > 0, T_d^2 = i\}.$$

*Proof.* In the following proof for simplicity we denote $\tilde{\boldsymbol{x}}_{S_1} = \boldsymbol{x} \odot T^1 \odot T^2; \tilde{\boldsymbol{x}}_{S_2} = \boldsymbol{x} \odot T^1 \odot (1 - T^2)$.

**Step 1.** Given $f, g, h$ projects input distributions into joint Gaussian distributions with fixed dimensionality, by convergence of Gaussian covariance matrices, we have:

$$\hat{\Sigma}(f(\tilde{\boldsymbol{x}}_{S_1}^i), h(\tilde{\boldsymbol{x}}_{S_2}^{N(i)})) = \frac{1}{n} \sum_{i=1}^n [f(\tilde{\boldsymbol{x}}_{S_1}^i); h(\tilde{\boldsymbol{x}}_{S_2}^{N(i)})][f(\tilde{\boldsymbol{x}}_{S_1}^i); h(\tilde{\boldsymbol{x}}_{S_2}^{N(i)})]^T \xrightarrow{a.s.} \Sigma_{f(\tilde{\boldsymbol{x}}_{S_1}^i), h(\tilde{\boldsymbol{x}}_{S_2}^{N(i)})};$$

$$\hat{\Sigma}(f(\tilde{\boldsymbol{x}}_{S_1}^i), g(\tilde{\boldsymbol{x}}_{S_2}^i)) = \frac{1}{n} \sum_{i=1}^n [f(\tilde{\boldsymbol{x}}_{S_1}^i); g(\tilde{\boldsymbol{x}}_{S_2}^i)][f(\tilde{\boldsymbol{x}}_{S_1}^i); g(\tilde{\boldsymbol{x}}_{S_2}^i)]^T \xrightarrow{a.s.} \Sigma_{f(\tilde{\boldsymbol{x}}_{S_1}^i), g(\tilde{\boldsymbol{x}}_{S_2}^i)}. \tag{21}$$

As in the Gaussian case, the mutual information between jointly Gaussian r.v.s is a function of the covariance matrix, we have

$$\hat{I}(f(\tilde{\boldsymbol{x}}_{S_1}^i); h(\tilde{\boldsymbol{x}}_{S_2}^{N(i)})) \xrightarrow{a.s.} I(f(\tilde{\boldsymbol{x}}_{S_1}^i); h(\tilde{\boldsymbol{x}}_{S_2}^{N(i)})) = I(\tilde{\boldsymbol{x}}_{S_1}^i; \tilde{\boldsymbol{x}}_{S_2}^{N(i)});$$
$$\hat{I}(f(\tilde{\boldsymbol{x}}_{S_1}^i); g(\tilde{\boldsymbol{x}}_{S_2}^i)) \xrightarrow{a.s.} I(f(\tilde{\boldsymbol{x}}_{S_1}^i); g(\tilde{\boldsymbol{x}}_{S_2}^i)) = I(\tilde{\boldsymbol{x}}_{S_1}^i; \tilde{\boldsymbol{x}}_{S_2}^i); \tag{22}$$
$$\mathbb{P}(\lim_{N\to\infty} \text{Empirical mTE} = \text{mTE}) = 1.$$

**Step 2.** Importantly, in our formulation eq (5), the $T_1, T_2$ are sampled once in one epoch, meaning they are fixed across features for computing mTE. Further note that $\sum_{d=1}^{p} \mathbb{P}(T_d^1 > 0) = \mathbb{E}||T^1||_0$; $\sum_{d=1}^{p} \mathbb{P}(T_d^2 \in (0,1)) = \mathbb{E}||\mathbb{1}_{T^2 \in (0,1)}||_0$. This means denoting the value of eq (5) as $L$, we have

$$L \xrightarrow{a.s.} \mathbb{E}_{T^1, T^2}[-\text{mTE}(\mathbb{1}_{T^1 \odot T^2 > 0}, \mathbb{1}_{T^1 \odot (1-T^2) > 0}) + \lambda_1||T^1||_0 + \lambda_2||\mathbb{1}_{T^2 \in (0,1)}||_0]$$
$$\geq \min_{T^1, T^2} -\text{mTE}(\mathbb{1}_{T^1 \odot T^2 > 0}, \mathbb{1}_{T^1 \odot (1-T^2) > 0}) + \lambda_1||T^1||_0 + \lambda_2||\mathbb{1}_{T^2 \in (0,1)}||_0. \tag{23}$$

Note with step 1 of the proof of theorem 2.4, for any $T_1$ we have

$$- \text{mTE}(\mathbb{1}_{T^1 \odot T^2 > 0}, \mathbb{1}_{T^1 \odot (1-T^2) > 0}) + \lambda_1||T^1||_0 + \lambda_2||\mathbb{1}_{T^2 \in (0,1)}||_0$$
$$\geq -\text{mTE}(\mathbb{1}_{T^1 \odot T^2 > 0}, \mathbb{1}_{T^1 \odot (1-T^2) > 0}) + \lambda_1||T^1||_0, \tag{24}$$

which is taken when $\forall d, \mathbb{P}(T_d^2 = 1) = 0/1, \mathbb{P}(T_d^2 = 0) = 1/0$. In this case,

$$||T^1||_0 = ||T^1 \odot T^2||_0 + ||T^1 \odot (1 - T^2)||_0.$$

Applying theorem 3.1, we have for $\lambda_1 = \lambda$ in theorem 3.1,

$$\min_{T^1, T^2} -\text{mTE}(\mathbb{1}_{T^1 \odot T^2 > 0}, \mathbb{1}_{T^1 \odot (1-T^2) > 0}) + \lambda_1||T^1||_0 + \lambda_2||\mathbb{1}_{T^2 \in (0,1)}||_0$$
$$= -\text{mTE}(S_1^*, S_2^*) + \lambda|S_1^* \cup S_2^*| := L^*. \tag{25}$$

Here $(S_1^*, S_2^*)$ satisfies properties described by theorem 3.1. Note the minimizer may not be unique, denote the set containing all minimizers as $\{(S_1^*, S_2^*)\}$. Then the equal sign in eq (23) holds if and only if $\mathbb{P}((\mathbb{1}_{T^1 \odot T^2 > 0}, \mathbb{1}_{T^1 \odot (1-T^2) > 0}) \in \{(S_1^*, S_2^*)\}) = 1$. Further noting $\forall d, \mathbb{P}(T_d^2 = 1) = 0/1, \mathbb{P}(T_d^2 = 0) = 1/0$, and our analysis above holds as $n \to \infty$ with probability 1 by a.s. convergence, we finally have

$$\mathbb{P}(\lim_{N\to\infty} \mathbb{P}(B_1 \subseteq S) = 1) = 1; \quad \mathbb{P}(\lim_{N\to\infty} \mathbb{P}(B_0 \subseteq S) = 1) = 1$$

holds. $\qquad\square$

# B  GATE INITIALIZATION

Our proposed initialization scheme is based on analysis of the linear case. Assume

$$f(X_{S_1}) = Xa, g(X_{S_2}) = Xb,$$

where $a, b \in \mathbb{R}^p$ represents two feature loadings. Then:

1. $a, b$ should be non-overlapping, therefore we expect $|a^T b|$ to be small.

2. We should have $f(X) \approx Wg(X)$ to maximize the mTE.

The constraint can be formulated into a regression problem $WXb = Xa$, therefore a natural solution is given by $a = X^\dagger WXb = (X^T X)^{-1} X^T WXb$. In this case, $||a^T b|| = ||b^T (X^T X)^{-1} X^T WXb|| = ||b||^2_{(X^T X)^{-1} X^T WX}$. Given $b$ is normalized, it can be shown that the optimal $b$ corresponds to the eigenvector with least absolute eigenvalue of matrix $(X^T X)^{-1} X^T WX$.

After getting $a, b$, we select a quantile threshold over $a/(a + b)$ to initialize the second stochastic gate layer. The first stochastic gate layer is initialized with uniform weights.

# C EXPERIMENTAL DETAILS

## C.1 TIME-SERIES BENCHMARKING STUDY

In the study the causal processes is simulated with Python package `Tigramite`. Among the total 100 features, there are 6 interacting features $\{1, 2, 3, 4, 5, 6\}$. The causal links are: 1->2 with time lag 2, 2->3 with time lag 1, 5->4 with time lag 1, 1->5 with time lag 1, 3->6 with time lag 3. These features also have autocorrelations with time lags ranging from 1 to 3. There is also a latent confounder modeled by `Tigramite` interacting with feature 0 and feature 2. In the case of strong latent process, the latent confounder also have effects on other 43 features. All other features (93/50) not mentioned above are nuisance features with white noise dynamics. The forward operator is defined by 5-neighbor lower triangular matrix.

### C.1.1 ALGORITHM IMPLEMENTATION

If not particularly mentioned, default settings of the algorithms are used throughout.

- **VAR-LINGAM.** The VAR-LINGAM algorithm is implemented in the Python package `LINGAM`, available at https://github.com/cdt15/lingam. VAR-LINGAM gives a weighted matrix as output. Therefore in our benchmarking study, we choose the most significant edge corresponding features with the number matching the sparsity level.

- **PCMCI.** The PCMCI algorithm is implemented in the Python package `Tigramite`, which gives a weighted matrix as output. We choose the most significant edge corresponding features with the number matching the sparsity level.

- **GVAR.** The GVAR algorithm is implemented at https://github.com/i6092467/GVAR. The sparsity parameter is set to be 1. We use the stable training option in GVAR, which trains the first and second half of the time series respectively to optimize over edge selection sparsity level then train on the whole time series, giving a binary output and no threshold selection is needed.

- **Grid-net.** The Grid-net algorithm is implemented at https://github.com/alexw16/gridnet. The parameter set: order=5, hidden_layer_size = 10, end_epoch=50, batch_size = 50, lmbd=1 is used throughout our study. After the training finishes, we choose the most significant edge corresponding features with the number matching the sparsity level.

- **DCM, NGM.** The two algorithms are both implemented at https://github.com/alexisbellot/ Graphical-modelling-continuous-time. For DCM, the default setting is used, and we use hidden dim = 10 for NGM. After both training finishes, we choose the most significant edge corresponding features with the number matching the sparsity level.

- **GEASS.** We use the same training parameters in all time-series settings, with the key sparsity regularization parameter $\lambda_1$ set with 0.04/0.05 based on a validation set, and the rest parameter settings are consistent with default.

### C.1.2 SCALABILITY ANALYSIS

We test PCMCI, GVAR, GrID-net, NGM, GEASS, GEASS+LPCMCI's running time with consistent settings described in the above section. (LPCMCI's setting is consistent with PCMCI's setting). We use the same data generation pipeline and select the set of the total feature numbers as $[100, 200, 400, 800, 1600]$.

## C.2 SIMULATED SPATIAL OMICS DATA BENCHMARKING STUDY

In the study the spatial omics data is simulated with Python package `Scsim` (Kotliar et al., 2019). 1000 genes are simulated in total, while 990 genes are cell-type-specificly expressed. The rest 10 genes each has a functional relationship (linear/nonlinear) with one cell-type-specific genes plus the noise term in order to model the cell-type-specific interactions. The data is then normalized and log-transformed according to the standard Scanpy pipeline (Wolf et al., 2018). The forward operator is defined by 4-neighbor adjacency matrix.

### C.2.1 ALGORITHM IMPLEMENTATION

If not particularly mentioned, default settings of the algorithms are used throughout.

- **Lasso Granger.** The Lasso algorithm is implemented by Scipy with tuned $\alpha$ (0.12) to match the sparsity level.
- **NCEM.** NCEM (Linear) is a linear graph neural network, which in the grid case corresponds to a standard linear regression based on neighbors and the cell type label. Based on the original work, we implemented our equivalent version by Lasso regression with $\alpha = 0.019$ to match the sparsity level.
- **GEASS.** We use the same training parameters in all settings, with the key sparsity regularization parameter $\lambda_1$ set with 0.02 based on a validation set, and the latent dimension number is set to be 64.
- **TE.** To give a fair comparison, we use the same architecture as GEASS except for the loss function is changed. We use the same training parameters in all time-series settings, with the key sparsity regularization parameter $\lambda_1$ set with 0.05 based on a validation set, and the latent dimension number is set to be 64 consistent with GEASS.

### C.3 SCRNA-SEQ PANCREAS TRAJECTORY

The data preprocessing is consistent with the scVelo tutorial: https://scvelo.readthedocs.io/ VelocityBasics/ (Bergen et al., 2020). The parameter set: $\lambda_1 = 0.06, \lambda_2 = 0.1$. Here because the gene regulatory network is fully connected and activated in cascade along the developmental trajectory, we consider the opposite initialization with $b$ be the largest eigenvalues corresponding eigenvectors of the matrix $(X^T X)^{-1} X^T W X$.

### C.4 MERFISH SPATIAL TRANSCRIPTOMICS DATA

The data is downloaded from Dryad and preprocessed with the standard Scanpy pipeline (Wolf et al., 2018): first normalize and log-transform the data by default functions in Scanpy then select 1000 highly variable genes by default functions in Scanpy (Wolf et al., 2018). The forward operator is defined by 5-neighbor adjacency matrix. The GEASS parameter set is consistent with those used in the spatial omics benchmarking.

## D ADDITIONAL EXPERIMENTAL RESULTS

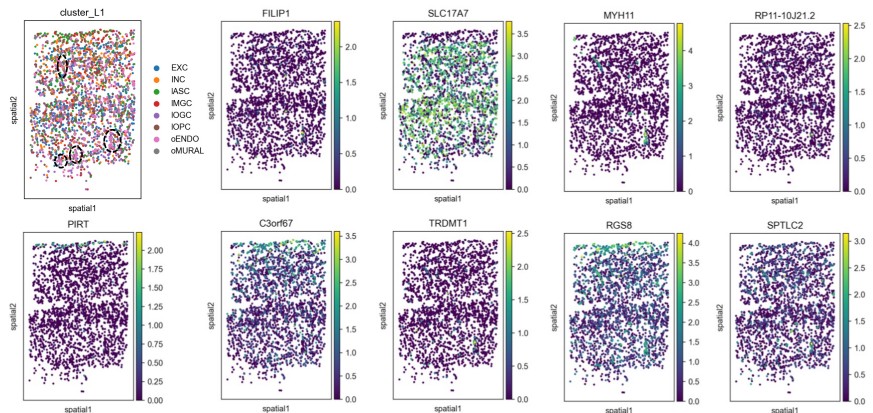

Figure 6: Spatial profiling of MERFISH human cortex slice, colored by cell type annotation and GEASS identified gene expressions. The ellipses in the first panel represent examples of vascular structures.

