# OpenReview forum: "GEASS: Neural causal feature selection for high-dimensional biological data"
_ICLR.cc/2023/Conference — ICLR 2023 notable top 25%_

### Official Review · Reviewer_WJXd · 2022-10-22

**Confidence:** 3
**Clarity, Quality, Novelty And Reproducibility:** The code is not provided. See above f…
**Correctness:** 4
**Technical Novelty And Significance:** 3
**Empirical Novelty And Significance:** 3
**Recommendation:** 8

**Strength And Weaknesses:**

Strengths: 1. This paper proposes a novel solution of causal feature selection problem in general graph-structured data.
                  2. Theoretical analysis is provided.
                  3. Empirical results showed outperformed performance.

Weaknesses: 1. I think the current version takes too much space on the theoretical analysis while ignoring the model itself. Figure 2 is not clear to me.
                      2. For the Merfish data experiment, what's the reason to downsample the gene number from 4000 to 1000? In real situations, the gene number can be millions. If the proposed method can not deal with high-dimensional features, then the scalability will be an issue.
                      3. What's the criteria for GEASS to select 9 genes in Merfish data?  Can the authors elaborate more /give insight on interpreting the selection of this subset for biological discovery?


**Summary Of The Paper:**

This paper identifies causally interacting features of high dimensional temporal/spatial data by considering the sparsity of underlying causal mechanisms instead of link sparsity, which can select the critical corresponding features for downstream causal discovery. Theoretical studies are provided, and empirical evaluations on synthetic and real biological data showed its superiority over competing methods.

**Summary Of The Review:**

I think this is a solid work.

---

> ### Author Response · Authors · 2022-11-18
> **Response to reviewer WJXd**
>
> We thank the reviewer for the supportive comments. The detailed response to each point is as follows. Please also see a summary of our major revisions above.
>
> **Regarding the model explanation**
>
> Thank you for bringing this up — some other reviewers have similar feedbacks for the manuscript. Therefore we have updated the manuscript with one additional section (3.2) to explain our proposed network architecture corresponding to Figure 2. Please let us know if there are further confusions.
>
> **Regarding the MERFISH experiment**
>
> **Gene number downsampling**
>
> Here we downsample the genes because selecting highly variable genes is common practice in the field of scRNA-seq data analysis as an essential preprocessing step. While we really appreciate the reviewer carefully reading the manuscript and bringing this up, the choice of gene number has little to do with the scalability concern. First, our result in the benchmarking section shows the superior performance of the method in terms of running time, which can be explained by the discussion in the main text. Second, a preliminary experiment shows the method can finish 100 epoches’ training in ~12 seconds on the whole dataset with a MAC M1 Pro **CPU**. Sorry for the confusion here and we have revised this in the new version of the manuscript.
>
> **Gene selection**
>
> We selected the same parameter setting as in our spatial benchmarking. To minimize the false positive rate, the sparsity regularization parameter is set to be high, which is the reason why only 9 genes are selected. We cover some biological insights in the main text about biological meanings of these genes. However, systematic identifications of regulatory relationships within these genes remain an issue: as far as we know, there are currently no satisfactory methods for such analysis in spatial data. Therefore it remains our future work to develop possible methods for further interpretation.
>
> **Regarding the code**
>
> Our implementation of the method is now attached in the supplementary material.
>
> Thanks again and we are happy to take any questions / further discussions.

---

> > ### Comment · Reviewer_WJXd · 2022-12-04
> > **Response to authors**
> >
> > Thank you for the explanations. I will keep my rate since it is already "accept".

---

### Official Review · Reviewer_nQe4 · 2022-10-24

**Confidence:** 3
**Correctness:** 4
**Technical Novelty And Significance:** 3
**Empirical Novelty And Significance:** 3
**Recommendation:** 8

**Clarity, Quality, Novelty And Reproducibility:**

The paper is clearly written, with high-quality exposition and proofs supporting each non-trivial assertion. The methods appear to be novel, but I am not very familiar with this subfield. The results are well-described, but no code repository is provided, so reproducibility is limited.

**Strength And Weaknesses:**

The strengths are as follows:

The paper is well-written and coherently organised.

It has clear novelty in the use of maximum transfer entropy.

All the non-trivial mathematical statements are supported by proofs in the appendix.

The weaknesses are as follows:

It is unclear how the forward operator translates to spatial settings; the authors vaguely mention something to do with diffusion, but that would not be applicable unless the diffusion trajectory could be reconstructed in time.

It is also unclear how the mutual information is computed (via approximation? based on frequencies in the data?).

Many proofs are hand-wavy; this may make them clear/obvious to experts, but for the general reader they should be made more explicit.

Some of the extensions of the method to specific datasets seem somewhat ad hoc, rather than being motivated by the same principles as the main method.

**Summary Of The Paper:**

The paper provides a novel methodology for identifying significant causal features in spatiotemporal data, with primary applications in biology (e.g. scRNA-Seq data). Its primary novelty lies in the judicious use of a transfer entropy characterisation of feature significance/relevance and the design of a combinatorial stochastic gate layer.

**Summary Of The Review:**

The paper is an advance over the current state of the art and provides interesting results in several biological applications.

---

> ### Author Response · Authors · 2022-11-18
> **Response to reviewer nQe4**
>
> We thank the reviewer for the supportive comments. The detailed response to each point is as follows. Please also see a summary of our major revisions above.
>
> **Regarding the forward operator formulation in spatial setting**
>
> For the sake of simplicity, we can assume that the spatial data can be modeled as a Markov random field with underlying normalized graph adjacency matrix A. In practice the graph can be constructed by spatial proximity. In the model, we have each sample (node) i is independent of the non-adjacent nodes, conditioning on its direct neighborhood members $N(i)=(A_i>0)$. In this case, the forward operator W is defined by $WX_i = X_{A_i>0}$. In practice, we take the average over the neighborhood, giving $\overline{X}_{A_i>0} = (AX)_i$.  Therefore, we can see W’s action on each sample is analogous with diffusion on the graph defined by A. Identification of causal relationships between features in each spatial location and neighboring location leads to further understanding of cell communication patterns for spatial transcriptomics data.
>
> **Regarding the mutual information computation**
>
> In our framework, we assume ergodicity and stationarity of the data. In time-series data, this means the empirical mutual information between $X_t$ (t=1:T-s) and  $Y_{t+s}$ (t=s+1:T) forms an estimate of $I(X_t,Y_{t+s})$, which is constant over t. Thus, the mutual information of time-series can be estimated.
>
> **Regarding the proof**
>
> Thank you for bringing this up. We provide more rigorous proofs in the revised version of the manuscript, which can be verified more easily. Please also see a summary of our major revisions above for more information.
>
> **Regarding the extended applications**
>
> Our main applications in real datasets include scRNA-seq developmental trajectory and MERFISH spatial omics data. For each type of data, identification of causal regulatory patterns within the data is a highly significant yet challenging task. Graph structures for each data can be defined by additional knowledge (RNA velocity & spatial location, please see “Regarding the forward operator formulation in spatial setting” for further explanation). Moreover, for both data, our underlying assumption of sparse interacting feature numbers is reasonable thus our main method (GEASS) can be adopted reasonably.
>
> **Regarding the code**
>
> Our implementation of the method is now attached in the supplementary material.
>
> Thanks again and we are happy to take any questions / further discussions.

---

### Official Review · Reviewer_uZbN · 2022-10-25

**Confidence:** 3
**Clarity, Quality, Novelty And Reproducibility:** It is quite well written.
**Correctness:** 3
**Technical Novelty And Significance:** 2
**Empirical Novelty And Significance:** 2
**Recommendation:** 6

**Strength And Weaknesses:**

The problem of detecting causal features in spatial transcriptomics is an important problem. The authors promote a nice framework and benchmark on a variety of datasets.

**Summary Of The Paper:**

The authors introduce a framework of causal discovery in data using two steps: a causal feature selection in a general graph structured data and causal graph identification. To do the former, the authors define a multi-dimensional transfer entropy maximization metric and device ways to optimise it. They test out the framework in single cell and spatial data.

**Summary Of The Review:**

The paper addresses an interesting problem. The main idea is neat and is benchmarked sufficiently on quite a lot of data. Thus I recommend acceptance.

---

> ### Author Response · Authors · 2022-11-18
> **Response to reviewer uZbN**
>
> We thank the reviewer for the supportive comments. Please also see the summary of our major revisions above.

---

### Official Review · Reviewer_Qga7 · 2022-11-01

**Confidence:** 4
**Correctness:** 4
**Technical Novelty And Significance:** 3
**Empirical Novelty And Significance:** Not applicable
**Recommendation:** 8

**Clarity, Quality, Novelty And Reproducibility:**

Clarity: The manuscript is clearly written, although a full understanding of the algorithm details would need a frequent reference to the appendix and cited literature.

Quality: The quality of the manuscript is good, with an extensive investigation of the algorithm design, proof of the key theorems, and insights into the model development.

Reproducibility: The manuscript did not provide any code repositories associated with the model. With the current description of the methodology, the work can be potentially reproduced, yet not guaranteed.



**Strength And Weaknesses:**

Strength: The methodology involved in the optimized feature subset selection for causal discovery (Theorem 3.1, 3.4) and the design of the stochastic gate-based approach is important to the field of data mining. Experiment results show superior performance on both synthetic and real data, further validating the effectiveness of the model.

Weakness: First of all, in section 3.2, the author mentioned that “GEASS provides both outputs of active features and embeddings produced by causally interacting features. In this paper, we emphasize the use of the former as the latter embedding output may be complex and nonlinear, potentially requiring additional architectures to maximize its interpretability.” The statement is confusing as it did not specify where the “embeddings produced by causally interacting features” were obtained: whether they were from the stochastic gates, or the MLPs indicated in Fig. 2? Further, there are no explanations for the design and purpose of the MLPs in Fig. 2.  Also, the term “STG layer” in Fig. 2 has never been mentioned throughout the manuscript. The reviewer guesses it is referring to the stochastic gates but is not sure about it.


**Summary Of The Paper:**

The manuscript proposed a feature selection model for the causal discovery task based on the optimization of multi-dimensional transfer entropy of the input data by combinatorial stochastic gates. The proposed model was evaluated on by synthetic time series datasets with the presence of latent process, synthetic spatial progression of the scRNA-seq data, as well as the real (pancreatic endocrinogenesis trajectory) data. Experiment results indicate superior performance of the model over both classic and recent causal discovery methods.

**Summary Of The Review:**

A causal discovery framework with fundamental and important algorithmic development (Theorem 3.1 and 3.4), with good practical value as well.

---

> ### Author Response · Authors · 2022-11-18
> **Response to reviewer Qga7**
>
> We thank the reviewer for the supportive comments. The detailed response to each point is as follows. Please also see a summary of our major revisions above.
>
> **Regarding the content confusion**
>
> Thank you for pointing out the possible multiple confusions in section 3. These points are further addressed in the revised version of the manuscript (See section 3.2). For the specific confusions the reviewer have mentioned:
>
> 1. “**whether they were from the stochastic gates, or the MLPs indicated in Fig. 2? “**
>
> &nbsp;&nbsp; The embeddings are obtained by MLPs indicated in Fig. 2.
>
> 2. “**Also, the term “STG layer” in Fig. 2 has never been mentioned… guesses it is referring to the stochastic gates but is not sure about it.”**
>
> &nbsp;&nbsp; Yes, the “STG” refers to stochastic gate layer.
>
> Further explanations can be seen in the response to the following point about MLP design.
>
> **Regarding the design and purpose of the MLPs**
>
> Ideally, given the result in section 2 of the paper, we can directly optimize the mTE function to find the sets $S_1$ and $S_2$ with the smallest union set size. However, because the original distribution of features can be arbitrary, the mutual information terms cannot be explicitly defined. In this work, we take an alternative approach by transforming the input selected feature distributions to (joint) Gaussian distributions. The rationale is that the mutual information between (joint) Gaussian distributions can be explicitly derived as functions of sample covariance matrices. The transform can be defined as neural networks of various architectures. In this work, we choose MLPs because of its simplicity, while we note other architectures can also be potentially adopted. The Gaussianity of the distributions can be enforced by adding a cross-entropy penalty with a Gaussian distribution of learnable means and variances.
>
> **Regarding the code**
>
> Our implementation of the method is now attached in the supplementary material.
>
> Thanks again and we are happy to take any questions / further discussions.

---

### Author Response · Authors · 2022-11-18
**To all reviewers: Summary of major revisions**

We sincerely thank all reviewers for their comments. We have substantially revised the manuscript to address each reviewer’s feedback. We summarize the major revisions here for reference:

1. A majority of proofs and assumptions are rewritten in a more mathematically rigorous manner, making the manuscript (especially the proof part) more accessible to general readers.
2. We merge our theory developed for bipartite causal graphs and general DAG causal graphs as there is large redundancy. We use the resulting additional space to add a section describing the network architecture shown in Figure 2, regarding the general feedback from the reviewers.
3. We reformulate the assumption 6 and discuss the meaning of the assumption in the context of conditional transfer entropy, provided in Appendix A.2.

Also, our implementation of the method is now attached in the supplementary material.

---

### Decision · Program_Chairs · 2023-01-20

**Decision:**

Accept: notable-top-25%

**Justification For Why Not Higher Score:**

There could be limitations in its applicability in other biological data sets beyond single cell rna, due to the spatiotemporal nature of the technique

**Justification For Why Not Lower Score:**

The method is novel with theoretical guarantees and practical impact

**Metareview: Summary, Strengths And Weaknesses:**

All reviewers agree that the paper offers a novel practical method for extracting causal features from high dimensional spatiotemporal data, based on granger causality; wtih applications in biology.

**Note From Pc:**

if the above contains the word "oral" or "spotlight" please see: "oral" presentation means -> notable-top-5% and "spotlight" means -> notable-top-25%. As stated in our emails, we are disassociating presentation type from AC recommendations